# The MAMA Algorithm for Fast Computations of Upwelling Far- and Mid-Infrared Radiances in the Presence of Clouds

Michele Martinazzo [†] and Tiziano Maestri *,[†]

Physics and Astronomy Department "Augusto Righi", University of Bologna, I-40127 Bologna, Italy; michele.martinazzo2@unibo.it

**\*** Correspondence: tiziano.maestri@unibo.it

**†** These authors contributed equally to this work.

**Abstract:** A methodology for the computation of spectrally resolved upwelling radiances in the presence of atmospheric diffusive layers is presented. The algorithm, called MAMA (Martinazzo–Maestri), provides fast simulations over the whole longwave spectrum, with high accuracy, particularly for optically thin scattering layers like cirrus clouds. The solution is obtained through a simplification of the multiple-scattering term in the general equation of the radiative transfer in a plane-parallel assumption. The scattering contribution is interpreted as a linear combination of the mean ambient radiances involved in the forward and back-scatter processes, which are multiplied by factors derived from the diffusive features of the layer. For this purpose, a fundamental property of the layer is introduced, named the angular back-scattering coefficient, which describes the fraction of radiation coming from a hemisphere and back-scattered into a specific direction (the observer in our case). This property, easily derived from the phase function of the particle size distribution, can be calculated from any generic single-scattering properties database, which allows for simple upgrades of the reference optical properties within the code. The paper discusses the solutions for mean upward and downward ambient radiances and their use in the simplification of the general radiative transfer equation for thermal infrared. To assess the algorithm performance, the results obtained with the MAMA code are compared with those derived with a discrete ordinate-based radiative transfer model for a large range of physical and optical properties of ice and liquid water clouds and for multiple atmospheric conditions. It is demonstrated that, for liquid water clouds, the MAMA code accuracy is mostly within $0.4 \ \mathrm{mW}/(\mathrm{m}^2\mathrm{cm}^{-1}\mathrm{sr})$ with respect to the reference code both at far- and mid-infrared wavelengths. Ice cloud spectra are also accurately simulated at mid-infrared for all realistic cloud cases, which makes the MAMA code suitable for the analysis of any spectral measurements of current satellite infrared sounders. At far infrared, the MAMA accuracy is excellent when ice clouds with an optical depth of less than 2 are considered, which is particularly valuable since cirrus clouds are one of the main targets of the future mission FORUM (Far-infrared Outgoing Radiation Understanding and Monitoring) of the European Space Agency. In summary, the MAMA method allows computations of cloudy sky high-resolution radiances over the full longwave spectrum (4–100 µm) in less than a second (for pre-computed gas optical depths and on a standard personal computer). The algorithm exploits the fundamental properties of the scattering layers, and the code can be easily updated in relation to new scattering properties.

**Keywords:** radiative transfer; fast models; multiple scattering; scaling methods; far infrared

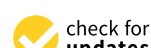



## 1. Introduction

In the last two decades, the scientific community has increased its involvement in the study of the Earth emission at far infrared (FIR) wavenumbers (approximately 100–677 cm$^{-1}$). The growing interest in this part of the spectrum is justified by the important role played by FIR radiation in shaping the Earth's energy balance and by its sensitivity to essential climate variables, such as temperature, water vapor, surface emissivity and

clouds. At the terrestrial temperature, roughly half of the Earth's total energy emission occurs at FIR wavelengths. In this spectral region, water vapor exhibits important absorption features, making the FIR highly valuable for improving the characterization of the atmospheric water vapor content and profile [1]. In conditions of low humidity, such as those encountered at high latitudes or at elevated locations, the water vapor absorption at FIR is reduced, thus allowing surface emitted radiation to reach the sky. This opens the possibility to derive the FIR surface emissivity from satellite observations [2,3] and possibly reduce uncertainties in climate models related to wrong assumptions about ice and snow emission properties (i.e., [4]). Radiation fields at FIR wavelengths are also strongly influenced by the presence of clouds. In particular, recent studies have demonstrated the larger sensitivity of FIR radiance to scattering by ice particles with respect to the mid-infrared part of the spectrum [5–7]. Despite its significance, mostly due to technical difficulties, the FIR remains currently unobserved from space. To address this observational gap, both the European Space Agency (ESA) and the National Aeronautics and Space Administration (NASA) are running innovative satellite projects which aim at measuring the FIR component at different spectral resolutions. Specifically, ESA, in 2019, selected the Far-infrared Outgoing Radiation Understanding and Monitoring (FORUM) mission as its ninth Earth Explorer, scheduled to launch in 2027 [8]. FORUM will collect measurements of the outgoing longwave radiation in the spectral range that goes from 100 to 1600 cm$^{-1}$, with 0.5 cm$^{-1}$ (unapodized) spectral resolution. On its side, NASA will launch, in 2024, two CubeSats loading the Polar Radiant Energy in the Far-InfraRed Experiment (PREFIRE) [9], which will measure the 0–54 μm region at 0.84 μm spectral resolution.

In this context, the scientific community involved in the FORUM and PREFIRE missions is currently assessing the ability of fast radiative transfer codes to simulate radiance fields at FIR wavelengths in order to set up the appropriate algorithms for the definition of the level 2 products and the analysis of the future measurements that will be performed by the sensors of the two aforementioned missions. The method used to solve the radiative transfer equation has a significant impact on the accuracy and speed of the simulation. In a clear sky scenario, as long as the thermal infrared region is considered (say 4 to 100 μm), radiative transfer in the atmosphere is dominated by absorption and emission processes. In this case, the solution is straightforward, and once the gaseous optical depths are known, the simulation run can be very fast. On the other hand, when clouds or aerosols are considered, the radiative transfer solution becomes complicated due to multiple-scattering processes. In the infrared part of the spectrum, the scattering effects cannot be neglected, and they are particularly important at FIR. Consider, for instance, that a maximum in diffusion processes for ice clouds is obtained at around 410 cm$^{-1}$, where the imaginary part of the refractive index of ice reaches a local minimum, and, consequently, a minimum in the absorption of ice crystals is observed ([10]).

In the last half century, a large number of numerical methods (e.g., doubling-adding [11,12], discrete ordinate methods [13], Monte Carlo method [14] or the successive-orders-of-scattering approach [15]) have been developed to solve the radiative transfer equation in the presence of scattering layers, such as clouds and aerosols. These "reference" methods have been used to extensively validate the single-scattering properties database against measurements in the FIR and MIR parts of the spectrum (i.e., [16–19]). Among the various methods, one of the most accepted numerical schemes for multiple scattering is DISORT (based on the discrete ordinate radiative transfer [20]), which can be run in chain with the popular code LBLRTM (line-by-line radiative transfer model [21]), which is used for the computation of high spectral resolution gaseous optical depths. A LBLRTM-DISORT wrapper codes is called LBLDIS [22]. LBLDIS exploits the discrete ordinate method to handle the multiple-scattering problem and resolve the radiance field in a plane-parallel atmosphere. In this work, LBLDIS is taken as the reference radiative transfer solution.

Nevertheless, radiative transfer algorithms laying on numerical schemes that rigorously solve the multiple-scattering equation require significant computational times; thus, they are unsuitable in numerical weather prediction data assimilation routines [23],

the analysis of a large dataset, or fast operational retrievals of geophysical parameters and atmospheric composition (e.g., [24,25]). Hyper-fast methodologies exist, capable of computing high-resolution radiances over the full-infrared spectrum in less than 0.05 s on a standard personal computer. An example is PCRTM [26], which exploits the principal component analysis to reconstruct the entire spectra given few computations on assigned monochromatic channels. Usually, this class of model relies on pre-computed lookup tables and routines, making it difficult to incorporate new releases of optical properties without remarkable computational efforts. In addition, an explicit treatment of the micro-physics is generally missing.

In this framework, fast, analytical and approximate methods are desirable. The two- and four-stream approximations are important examples of these methodologies, widely applied to perform fast irradiance and radiance computations [27–29] in the time of the order of few seconds on a normal personal computer. To further reduce the computational time, one can utilize a scaling approach, which allows to simplify the radiative transfer equation by avoiding the direct computation of the multiple-scattering scheme. A notable example is the Chou approximation [30]. In this scheme, the multiple-scattering effects are accounted for by scaling the optical depth of the layer, reducing the radiative transfer equation to a Schwarzschild-like equation. The radiance field is thus described by assuming a plane-parallel atmosphere in the local thermodynamic equilibrium and for emission and absorption processes only. These approximations yield an extremely simple solution, which can be implemented in models originally running in clear sky conditions only, without affecting the modeling structure of the code. Nevertheless, assumptions made to apply scaling methods introduce errors in the simulation, which results in a general overestimation of the computed fluxes and radiances as assessed in the works by [31–33]. In the last years, efforts have been made to enhance the accuracy of these scaling methodologies. An attempt in this regard is the adjustment scheme proposed by [31] for the computation of the long-wave upwelling irradiances. The adjustment algorithm was recently adapted to the computation of radiances by [34], which show that accurate solutions can be obtained, including at FIR wavelengths, in the presence of thin ice clouds (for OD of less about unity) and for any combination of effective radius and atmospheric conditions. One drawback of the scaling methods for the computation of the radiance fields, also when in combination with adjustment algorithms, is the necessity to account for a corrective coefficient, which usually requires a large set of numerical simulations and the use of a reference model for its definition. In [34], it is shown that the correctional coefficient can be parameterized in terms of effective radius and observational angle.

This work presents a new analytical approach, called MAMA (Martinazzo–Maestri), for the fast computation of longwave (MIR and FIR) radiance in a plane-parallel assumption. The method is capable of performing accurate simulations of the upwelling radiances (at nadir) in the presence of optically thin scattering layers such as cirrus clouds with OD (at 900 cm$^{-1}$) less than 2. The method offers the advantage of relaying the fundamental properties of the scattering layers (such as clouds and aerosols) and accounting for an explicit treatment of their micro-physical properties. MAMA does not require the computation of any adjustment coefficient (or other calibration methodology), and it is easy to expand or update in case new releases of the single-scattering properties of aerosols and clouds are made available.

The paper is organized as follows: In Section 2, the radiative transfer equation and the rationale of the MAMA approximation are presented. Two subsections introduce the methodology used for the computations of the mean upward and downward radiation. The last subsection of Section 2 provides the approximate final radiative transfer equation and its solution. In Section 3, the assessment of the accuracy of the MAMA algorithm is performed by comparing it against a reference solution and for multiple conditions of atmospheric state and cloud properties. Conclusions are drawn in Section 4. Three Appendix sections provide important information. Appendix A provides the estimate of the radiance parallel to the surface within the cloud which is critical for the definition of

the mean upwelling radiance. In Appendix B, the newly introduced optical properties of the scattering layer are described. Appendix C discusses the generalization of the MAMA solution to off-nadir observational angles.

## 2. The Radiative Transfer Equation and the MAMA Approximation

In a plane-parallel approximation with azimuthal symmetry, the radiative transfer equation in local thermodynamic equilibrium describing the upward monochromatic radiance $I$ at longwave wavelengths and in the presence of multiple-scattering events is

$$\mu \frac{dI(\tau, \mu)}{d\tau} = I(\tau, \mu) - [1 - \tilde{\omega}(\tau)]B(\tau) - \frac{\tilde{\omega}(\tau)}{2} \int_{-1}^{1} d\mu' I(\tau, \mu') P(\mu, \mu') \tag{1}$$

where $\mu = \cos(\theta)$ indicates the direction of observation of the monochromatic radiance $I$, $\tau$ is the integrated optical depth from the top of the atmosphere (TOA) to the level of interest, $\tilde{\omega}$ is the single-scattering albedo of the layer at level $\tau$, $B$ is the Planck function at level $\tau$, and $P(\mu, \mu')$ is the azimuthally averaged scattering phase function describing the scattering events for radiation entering the layer at zenith angle $\mu'$ and exiting in the $\mu$ direction.

An accurate description of the multiple-scattering processes (last term of Equation (1)) requires an explicit description of the angular dependence of the radiance field. This leads to an increase in the dimensionality of the problem, which becomes computationally demanding, especially when conducting simulations with high spectral resolution. For computations of cloudy sky spectra in the MIR part of the spectrum (about from 667 to 2500 cm$^{-1}$), scaling methods such as the one proposed by Chou [30] or other solutions based on the similarity principle [35] provide sufficiently accurate results as shown in [33], even if initially conceived for computations of spectral irradiance fluxes. The main reason lies in the fact that the spectral signatures of the radiance at atmospheric windows in the mid-infrared region are mostly dependent on the clouds absorption features than on scattering ones [36], which can thus be easily approximated by a correction term. Nevertheless, in the FIR part of the spectrum, and mainly around 410 cm$^{-1}$, the imaginary part of the index of refraction of ice reaches a local minimum, making the ice crystal absorption very low and thus enhancing the role of scattering [6]. It is, in fact, demonstrated that in the far-infrared portion of the spectrum, the sensitivity of radiance to the assumed ice habit is notably greater compared to mid-infrared wavelengths [8,37]. In this region, simple scaling methods, conceived for fast radiative fluxes computations, fail at reproducing the heterogeneous angular variation of the radiance field when multiple-scattering events are occurring [33].

To overcome this problem, a new methodology is presented, which aims at approximating the radiative transfer equation for the upwelling radiance, with the goal to allow fast but accurate computations in the presence of clouds and aerosol layers in the infrared part of the spectrum (100–2500 cm$^{-1}$). The description of the methodology accounts for a simplification of the radiative transfer equation. Following the scheme proposed by Tang et al. (2018) [31], the multiple-scattering integral term is separated into two distinct components, specifically the forward-scattering and backward-scattering contributions. These components correspond to the last two terms in the following Equation (2):

$$\mu \frac{dI(\tau, \mu)}{d\tau} = I(\tau, \mu) - [1 - \tilde{\omega}(\tau)]B(\tau) +$$
$$- \frac{\tilde{\omega}(\tau)}{2} \left[ \int_{-1}^{0} d\mu' I(\tau, \mu') P(\mu', \mu) + \int_{0}^{1} d\mu' I(\tau, \mu') P(\mu', \mu) \right] \tag{2}$$

From the above, two average quantities, representative of the upward and downward ambient radiances (respectively $\langle I_u(\tau) \rangle$ and $\langle I_d(\tau) \rangle$) are defined so that Equation (2) is written as

$$\mu \frac{dI(\tau, \mu)}{d\tau} = I(\tau, \mu) - [1 - \tilde{\omega}(\tau)]B(\tau) + \\ - \frac{\tilde{\omega}(\tau)}{2} \left[ \langle I_d(\tau) \rangle \int_{-1}^{0} d\mu' P(\mu', \mu) + \langle I_u(\tau) \rangle \int_{0}^{1} d\mu' P(\mu', \mu) \right] \tag{3}$$

Note that the integral terms in Equation (3) depend only on the layer-scattering properties. The values of these terms are completely determined once the single-scattering properties and the particle size distribution are assumed. It is thus possible to define a new optical parameter, here called the angular back-scattering coefficient $c(\mu)$, which describes the fraction of radiation coming from a hemisphere and back-scattered in the $\mu$ direction. Formally,

$$c(\mu) = \frac{1}{2} \int_{-1}^{0} d\mu' P(\mu', \mu) \tag{4}$$

And since the azimuthally averaged scattering phase function $P(\mu', \mu)$ is normalized, it turns out that

$$1 - c(\mu) = \frac{1}{2} \int_{0}^{1} d\mu' P(\mu', \mu) \tag{5}$$

which defines an angular forward-scattering coefficient.

The schematic representation of the angular back-scattering $c(\mu)$ and forward-scattering coefficient $1-c(\mu)$ in the zenith direction is provided in Figure 1. Note the conceptual similarity to the back-scatter parameter $b$, introduced by Chou, which was used to represent the fraction of radiation back-scattered over a full hemisphere and adopted in flux computations.

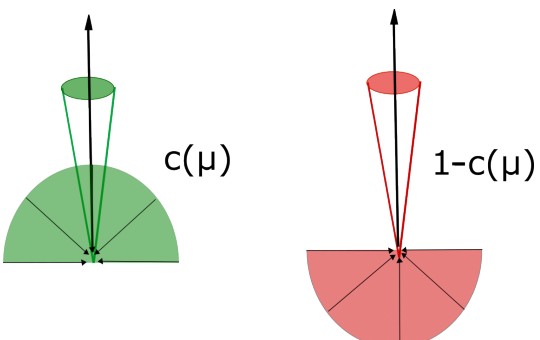

**Figure 1.** The angular back-scattering and forward scattering coefficients $1 - c(\mu)$ (red) and $c(\mu)$ (green). The semicircle indicates the hemispheric radiation scattered in the $\mu$ direction by the layer. In the figure $\mu = 1$.

To preserve the same structure as Equation (2), the averaged terms of Equation (3) must satisfy the following requirements:

$$\langle I_u(\tau, \mu) \rangle = \frac{\int_{0}^{1} d\mu' I(\tau, \mu') P(\mu', \mu)}{\int_{0}^{1} d\mu' P(\mu', \mu)} \tag{6}$$

and

$$\langle I_d(\tau, \mu) \rangle = \frac{\int_{-1}^{0} d\mu' I(\tau, \mu') P(\mu', \mu)}{\int_{-1}^{0} d\mu' P(\mu', \mu)} \tag{7}$$

These two quantities can be interpreted as effective radiances representative of the mean radiation involved in the forward- and back-scattering processes. The exact solutions of Equations (6) and (7) can only be derived when the angular dependence of the radiance

field is known. Nevertheless, we can employ certain assumptions to approximate these two terms. In the solution proposed by Chou [30], these two terms are assumed to be isotropic in both the upward and downward hemispheres, and equal to $\langle I_u(\tau) \rangle \approx I(\mu, \tau)$ and $\langle I_d(\tau) \rangle \approx B(\tau)$, respectively. With these assumptions, Equation (1) can be simplified, resulting in a Schwarzschild-like equation.

It has been proven that the application of the Chou scheme results in a systematic overestimation of simulated nadir radiances at the TOA [33]. These discrepancies can be primarily attributed to the average quantities employed to approximate Equations (6) and (7), which are larger than their true values. To establish a more accurate computational scheme for upward radiances, it is essential to incorporate a more realistic representation of these two terms.

In this work, we will focus on producing an improved solution for the radiance computation in the upwelling nadir-looking geometry, i.e., the solution of Equation (3) for $\mu = 1$. A more general solution encompassing other viewing angles is briefly discussed in Appendix C. By making use of the nadir value of the angular back-scatter and forward-scatter coefficients ($c(\mu = 1)$ and $1 - c(\mu = 1)$), Equation (3) becomes

$$\frac{dI(\tau,1)}{d\tau} = I(\tau,1) - [1 - \tilde{\omega}(\tau)]B(\tau) - \tilde{\omega}(\tau)[\langle I_d(\tau,1)\rangle c(1) + \langle I_u(\tau,1)\rangle(1 - c(1))] \quad (8)$$

where the observation zenith angle is set to $\mu = 1$ so that $c(\mu = 1) = c(1)$

### 2.1. Approximation for the Mean Upward Ambient Radiation $\langle I_u(\tau,\mu)\rangle$

The objective of this paragraph is to provide a solution for the mean upward radiance, shown in Equation (6), assuming a nadir-looking geometry ($\mu = 1$). An accurate representation of the upward ambient radiation $I(\tau, \mu')$ is required to obtain a satisfactory description of the multiple-scattering contribution. In the scaling method proposed by Chou et al. (1999) [30], this quantity is assumed to be constant over the entire hemisphere, which is not realistic for generic conditions. In fact, $I(\tau, \mu')$ usually decreases as the viewing zenith angle increases. In the present work, it is proposed to assume a simple relation between the ambient radiation $I(\tau, \mu')$ and the cosine of the observational angle $\mu'$. A linear relation (the dependency on the vertical coordinate is implicit) is considered:

$$I(\tau,\mu') = I^{\uparrow}(\tau) \cdot \mu' + \overrightarrow{I}(\tau) \cdot (1 - \mu'), \qquad \mu' > 0 \quad (9)$$

where $I^{\uparrow}(\tau)$ and $\overrightarrow{I}(\tau)$ represent two limit cases: the ambient radiation directed toward $\mu' = 1$ and toward $\mu' = 0$, respectively. The value for the first limit case ($\mu' = 1$) is simply the upward radiance $I(\tau, \mu = 1) = I(\tau, 1)$. On the other hand, when $\mu' = 0$, a proper description for $\overrightarrow{I}(\tau)$ needs to be found. For this purpose, the radiative transfer equation in the presence of multiple scattering, using the same assumption made by Chou et al. (1999), is exploited. With the assumption of the horizontal symmetry of the problem, the solution is

$$\overrightarrow{I}(\tau) = (1 - \frac{\tilde{\omega}}{2})B(\tau) + \frac{\tilde{\omega}}{2}I(\tau,1) \quad (10)$$

where $\tilde{\omega}$ is the single-scattering albedo of the layer. The detailed derivation of Equation (10) is described in Appendix A. Given these descriptions of the first and second limit cases ($\mu' = 1$ and $\mu' = 0$ respectively), the ambient radiance (Equation (9)) is written as

$$I(\tau,\mu') = I(\tau,1) \cdot \mu' + \left[(1 - \frac{\tilde{\omega}}{2})B(\tau) + \frac{\tilde{\omega}}{2}I(\tau,1)\right] \cdot (1 - \mu') \quad (11)$$

$I(\tau, \mu')$ in Equation (11) is thus used to explicitly solve Equation (6), describing the average upward radiation $\langle I_u(\tau, 1) \rangle$. By making use of the angular back-scattering coefficient, Equation (6) is written as follows:

$$\langle I_u(\tau, 1) \rangle = \frac{\frac{1}{2} \int_0^1 d\mu' \left[ I(\tau, 1) \cdot \mu' + \left[ (1 - \frac{\tilde{\omega}}{2})B + \frac{\tilde{\omega}}{2} I(\tau, 1) \right] \cdot (1 - \mu') \right] P(\mu', 1)}{1 - c(1)} \tag{12}$$

where $c(1) = c(\mu = 1)$ and $P(\mu', 1) = P(\mu', \mu = 1)$, as we are considering a nadir-looking case. After a few operations, the integral in the numerator of Equation (12) is easily derived:

$$\langle I_u(\tau, 1) \rangle \cdot [1 - c(1)] = \left[ \frac{\tilde{\omega}}{2} I(\tau, 1) + \left(1 - \frac{\tilde{\omega}}{2}\right) B(\tau) \right] \cdot \frac{1}{2} \int_0^1 d\mu' P(\mu', 1) + \\ \left[ \left(1 - \frac{\tilde{\omega}}{2}\right) I(\tau, 1) - \left(1 - \frac{\tilde{\omega}}{2}\right) B(\tau) \right] \cdot \frac{1}{2} \int_0^1 d\mu' P(\mu', 1) \mu' \tag{13}$$

Note that the first integral term in the numerator (Equation (13)) is $1 - c(\mu)$. The second integral term, on the other hand, is a new quantity, here called gamma, $\gamma(\mu)$:

$$\gamma(\mu) = \frac{1}{2} \int_0^1 d\mu' P(\mu', \mu) \mu' \tag{14}$$

This quantity $\gamma(\mu)$ resembles the asymmetry parameter, from which it differs only for the limits of integration. As the $c(\mu)$ parameter, it is a physical property of the layer that can be easily computed once the single-scattering properties and particle size distribution are assumed. If the exit direction is set to $\mu = 1$, the definition of the $\gamma(\mu)$ coefficient is equal to the definition of the asymmetry parameter $g$ over the forward hemisphere. For this reason, the value of $\gamma(\mu)$ gets closer to the value of $g$ as the effective dimension of the particle increases.

Using the definition (14), the average upward radiation becomes

$$\langle I_u(\tau, 1) \rangle = \frac{\left[ \frac{\tilde{\omega}}{2} I(\tau, 1) + \left(1 - \frac{\tilde{\omega}}{2}\right) B(\tau) \right] (1 - c(1)) + \left[ \left(1 - \frac{\tilde{\omega}}{2}\right) I(\tau, 1) - \left(1 - \frac{\tilde{\omega}}{2}\right) B(\tau) \right] \gamma(1)}{1 - c(1)} \tag{15}$$

Equation (15) will be used as the expression of the mean upward radiance to solve the radiative transfer Equation (3).

### 2.2. Approximation for the Mean Downward Ambient Radiation $\langle I_d(\tau, \mu) \rangle$

In this paragraph, a solution for the mean downward radiance term present in Equation (7) is provided for a nadir-looking geometry.

For a non-scattering atmosphere, the radiative transfer equation has a Schwarzschild-like form, and the computation of the downward and upward radiances is independent. This allows to approximate the average downward ambient radiation $\langle I_d(\tau, \mu) \rangle$ with a downward radiance computed at an optimal viewing angle $\tilde{\mu}$. In the proposed scheme, the scaling method derived by Chou et al. (1999) [30] is used for the calculation of the downward radiation, which will be described by an Equation (16) of the following form:

$$\mu \frac{dI(\tau, \mu)}{[1 - \tilde{\omega}(1 - b)]d\tau} = I(\tau, \mu) - B(\tau), \qquad \mu < 0 \tag{16}$$

where $b$ is the back-scattering coefficient as defined by Chou [30]. In the present context, a suitable description for the downward ambient radiation $I(\tau, \mu')$ is obtained by solving Equation (16) at an optimal observation angle $\tilde{\mu}$:

$$\tilde{\mu} \frac{dI(\tau, \tilde{\mu})}{[1 - \tilde{\omega}(1 - b)]d\tau} = I(\tau, \tilde{\mu}) - B(\tau) \tag{17}$$

It is assumed that $\tilde{\mu}$ is the cosine of an effective zenith angle $\theta = -60.0$ degrees ($\tilde{\mu} = 0.5$). The ambient downward radiation at level $\tau$ ($I(\tau, \mu')$) is therefore taken to be equal to the solution of Equation (17):

$$I(\tau, \mu') = I(\tau, \tilde{\mu}), \qquad \mu' < 0 \tag{18}$$

And Equation (7) becomes

$$\langle I_d(\tau, 1) \rangle = \frac{\int_{-1}^{0} d\mu' I(\tau, \tilde{\mu}) P(\mu', 1)}{\int_{-1}^{0} d\mu' P(\mu', 1)} = I(\tau, \tilde{\mu}) \tag{19}$$

Note that, for a single homogeneous layer, an analytical solution of Equation (17) can be obtained for $I(\tau, \tilde{\mu})$. Setting $\tau = 0$ at the top of the layer, it is derived that

$$\langle I_d(\tau, 1) \rangle = I(\tau, \tilde{\mu}) = I(0, \tilde{\mu}) e^{-\alpha_c \tau / \tilde{\mu}} + \left[1 - e^{-\alpha_c \tau / \tilde{\mu}}\right] B(\tau) \tag{20}$$

where $\alpha_c = 1 - \tilde{\omega}(1 - b)$ is the scaling term proposed by [30] , $I(0, \tilde{\mu})$ is the boundary condition at the top of the layer, and $B(\tau)$ is the Planck function, at level $\tau$, which is considered constant within the layer thickness.

Note that, in any radiative transfer model, the diffuse downward radiance needs to be computed to account for the surface reflection. For this reason, the computation of the mean downward ambient radiation comes at almost no cost.

### 2.3. Solution of the Radiative Transfer Equation

In the previous sections, the descriptions of the average upward and downward ambient radiation are obtained (respectively Equations (15) and (19)). These results are used to solve the equation for the upwelling radiance (Equation (8)), whose expression becomes

$$\frac{dI(\tau)}{d\tau} = I(\tau) - [1 - \tilde{\omega}(\tau)] B(\tau) +$$
$$- \tilde{\omega}\left[\left[\frac{\tilde{\omega}}{2} I(\tau) + \left(1 - \frac{\tilde{\omega}}{2}\right) B(\tau)\right](1 - c) + \left[\left(1 - \frac{\tilde{\omega}}{2}\right) I(\tau) - \left(1 - \frac{\tilde{\omega}}{2}\right) B(\tau)\right]\gamma\right] +$$
$$- \tilde{\omega} c I(\tau, \tilde{\mu}) \tag{21}$$

where the dependency on $\mu = 1$ (nadir view) is made implicit. After some algebra, Equation (21) is rearranged:

$$\frac{dI(\tau)}{d\tau} = \left[1 - \tilde{\omega}\gamma - \frac{\tilde{\omega}^2}{2}(1 - c - \gamma)\right] I(\tau) +$$
$$- \left[1 - \tilde{\omega}(c + \gamma) - \frac{\tilde{\omega}^2}{2}(1 - c - \gamma)\right] B(\tau) +$$
$$- \tilde{\omega} c I(\tau, \tilde{\mu}) \tag{22}$$

To improve its readability, the first coefficient on the right-hand side of Equation (22) is indicated as $\alpha$:

$$\alpha = 1 - \tilde{\omega}\gamma - \frac{\tilde{\omega}^2}{2}(1 - c - \gamma) \tag{23}$$

This allows to rewrite Equation (22) in a very simple form:

$$\frac{dI(\tau)}{d\tau} = \alpha I(\tau) - [\alpha - \tilde{\omega} c] B(\tau) - \tilde{\omega} c I(\tau, \tilde{\mu}) \tag{24}$$

This is the final form of the equation used by the MAMA algorithm to compute the upwelling radiance in a nadir-looking geometry.

This equation requires the computation of the fundamental optical properties of the scattering layers, such as the $c$, $\tilde{\omega}$ and $\gamma$ and the Chou back-scatter parameter $b$, which is hidden in the last term of Equation (24). The latter is solved by using Equation (17). For an in-depth analysis of the newly defined scattering parameters, see Appendix B.

Equation (24) allows for simple analytical solutions. Since the computation of the upward radiance can be made layer by layer, the focus is posed on single homogeneous layers. To simplify the problem, the temperature inside the layer can be assumed constant. This assumption is generally accurate only for sufficiently thin layers. However, the solution for layers where the temperature vary linearly with the vertical coordinate is also easy to compute.

Using (20) in (24), and solving for $I(\tau)$, the nadir-looking radiance at the top of the layer is derived as

$$I(\tau) = \left[1 - e^{-\alpha\tau}\right] B(\tau) + \left[1 - e^{-\left(\frac{\alpha_c}{\tilde{\mu}} + \alpha\right)\tau}\right] \frac{\tilde{\omega} c \left[I(0, \tilde{\mu}) - B(\tau)\right]}{\frac{\alpha_c}{\tilde{\mu}} + \alpha} \tag{25}$$

In conclusion, it is interesting to note that if we assume that the downward radiation is well described by the black body emission from the layer $I(\tau, \tilde{\mu}) = B(\tau)$ (as done in the Chou theory), Equation (24) simplifies into

$$\frac{dI(\tau)}{\alpha d\tau} = I(\tau) - B(\tau) \tag{26}$$

which can be interpreted as a new scaling method, where an apparent absorption optical depth is defined as $\tau_a = \alpha\tau$. The simplification has the advantage of being very easy to implement. Nevertheless, it produces a general overestimation in the upward radiance field at far- and mid-infrared wavelengths, such as Chou's solution (not shown). For this reason, the solution obtained from Equation (25) is the one considered in the MAMA algorithm.

## 3. Methods and Results

The results obtained from the comparison between the MAMA solution and the reference methodology are presented. The reference results are obtained from the DISORT (discrete ordinate radiative transfer model) [20] routine in the LBLDIS (LBLRTM + DISORT) [22] code chain. DISORT is a plane-parallel discrete ordinate algorithm for monochromatic unpolarized radiative transfer in non-isothermal, vertically inhomogeneous media. LBLDIS uses the optical depths computed by LBLRTM [38] and the DISORT v2.0 routine to solve the radiative transfer problem in the presence of multiple-scattering layers.

The vertical profiles of the concentration of the main gas species active at FIR and MIR wavelengths are obtained from the climatological database IG2 v5.7 [39], and are calculated in order to be representative of low-, mid-, and high-latitude atmospheres. The database is used for the specification of the pressure, temperature and gas volume mixing ratio for the following molecules (including isotopes): $H_2O$, $CO_2$, $O_3$, $N_2O$, $CO$, $CH_4$, $O_2$, $NO$, $SO_2$, $NO_2$, $NH_3$, and $HNO_3$. Figure 2 shows the vertical profiles of the temperature and water vapor mixing ratio, from the ground up to 20 km height, for the three scenarios considered, which are representative of low, medium and high latitudes.

For each one of the three scenarios reported in Figure 2, the presence of a single ice or liquid water cloud is assumed. Multiple conditions are considered, which account for different cloud heights, optical depths and microphysical parameters (i.e., the effective radius of the particle size distribution). A focus is placed on cirrus clouds and low-level water clouds. In case of medium–high clouds, the geometrical parameters (altitude of the height and thickness) are defined in accordance with the statistics found by [40], which is obtained by a global analysis of CALIOP data. The goal is to have an accurate characterization of the cirri, which is one of the goals of the FIR observations which will be performed by the FORUM mission. In the case of low-liquid water clouds, generic

features are assumed for layers below 3 km of altitude. The main parameters used for characterization of the clouds in the radiative transfer computations are briefly described in the following subsection.

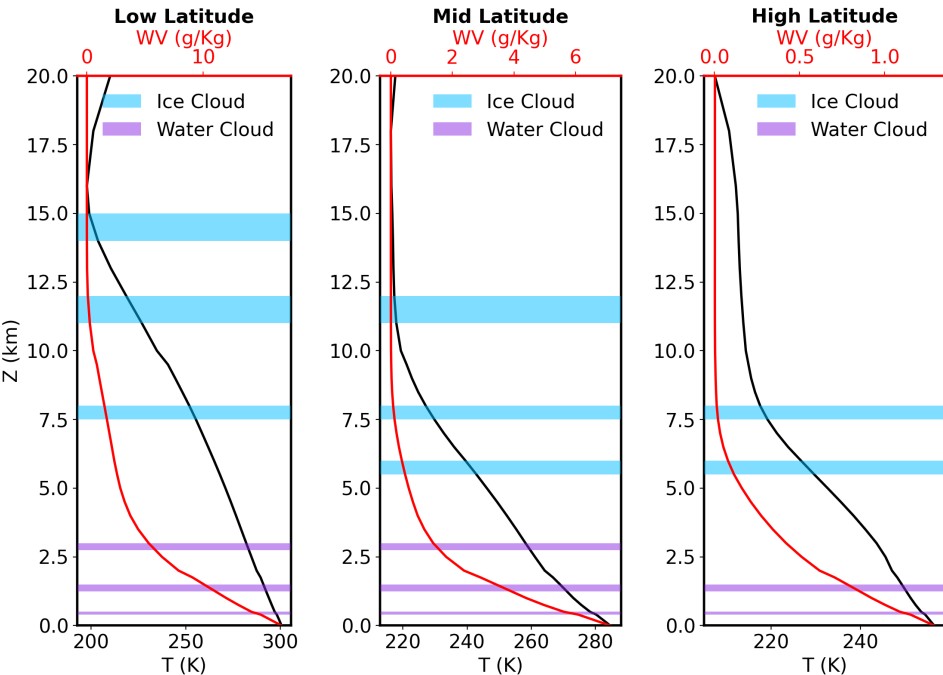

**Figure 2.** Temperature (black solid line) and water vapor mixing ratio (red line) vertical profiles for three different scenarios. Light blue and purple shaded bands indicate the height position and thickness of the ice and liquid water clouds, respectively, used in the computations.

### 3.1. Assumptions on Liquid Water and Ice Clouds

We consider liquid water clouds composed of water spheres, whose optical properties are generated by using a Mie solution-based algorithm. The Mie code is the Scattnlay model [41], which allows for the computation of scattering coefficients, efficiency factors, and scattering phase functions for single, isolated, spherical particles from complex refractive indices [42,43]. The optical properties are then combined to obtain the bulk radiative properties for the particle size distributions (PSDs) representative of the cloud layer, across the desired spectral range.

The analytical PSDs assumed to model the low-level water clouds are lognormal distributions [44], whose number of particles per unit volume is given by

$$n(r) = \frac{n_0}{r\sqrt{2\pi}\sigma} exp\left\{-\frac{(ln(r) - ln(r_m))^2}{2\sigma^2}\right\} \tag{27}$$

where $r$ is the particle radius, $r_m$ is the mode radius of the distribution, $\sigma$ is the scale parameter, and $n_0$ is a normalization factor depending on the total number of particles per volume used in the radiative transfer computations. According to [44], assuming low-level stratiform clouds, the average value for the scale parameter is set to $\sigma = 0.38$ and considered constant. This allows us to refer to different PSDs only by using their effective radius $r_{eff}$, which is defined as the fraction of the third to the second moment of the particle size distribution:

$$r_{eff} = \frac{\int_0^\infty r^3 n(r) dr}{\int_0^\infty r^2 n(r) dr} \tag{28}$$

It is easy to prove that for a lognormal PSD, the relation between effective and mode radius is governed by the following:

$$r_{eff} = r_m \cdot exp\left\{\frac{5}{2}\sigma^2\right\} \tag{29}$$

The simulations are performed considering different effective radii, optical depths (OD, at 900 cm$^{-1}$), and cloud heights. The equation defining the total OD for a vertically homogeneous cloud with a geometrical thickness $\Delta z$ is given by

$$OD = N_{tot} \cdot \beta(r_{eff}, 900) \cdot \Delta z \tag{30}$$

where $\beta(r_{\text{eff}}, 900)$ is the extinction coefficient at 900 cm$^{-1}$ of the PSD corresponding to a specific effective radius, normalized to a single particle per unit of volume. Finally, $N_{\text{tot}}$ represents the total number of particles in the considered volume.

Ice clouds observed in nature shows multiple crystal shapes, depending on the cloud thermodynamic conditions, formation processes and evolution. In this work, we refer to ice cloud as PSDs of aggregates of eight hexagonal ice columns, whose single-particle radiative properties are described by [6].

A commonly assumed PSD for ice clouds is the three-parameter gamma-type distribution, here below written as a function of $D$, which is the maximum dimension of the ice particle:

$$n(D) = n_0 D^{\mu} exp\{-\lambda D\} \tag{31}$$

where $n_0$ is the normalization factor, $\mu$ is the shape parameter, and $\lambda$ is the slope parameter. An average value of $\mu = 7$ is assumed in this work. For positive values of $\mu$, the shape of the gamma distribution is of the under-exponential type, and the maximum of the distribution lies in between the minimum and maximum dimensions of the crystals. As for the case of water clouds, assuming a constant value for $\mu$ allows us to describe the different PSDs using the effective dimension only. For non-spherical particles, an effective dimension of the distribution can be defined as

$$D_{eff} = \frac{3}{2}\frac{\int_0^{\infty} dD V(D)n(D)}{\int_0^{\infty} dD A(D)n(D)} = 2r_{eff} \tag{32}$$

where $A(D)$ and $V(D)$ are the cross-sectional area and the volume of the particle with maximum dimension $D$.

The atmospheric profiles and cloud properties described above are used as inputs for a wide range of simulations. Table 1 summarizes the main cloud parameters used in the simulations and their range of variation.

**Table 1.** The main cloud parameters (and their range) used for the simulation comparisons.

| Particle Type | PSD Type (Fixed Parameter) | $r_{\text{eff}}$ (µm) | Top Height (km) | OD (at 900 cm$^{-1}$) |
|---|---|---|---|---|
| Liquid water droplets | lognormal ($\sigma = 0.38$) | 1–20 | 0.5–3 | 1–50 |
| Ice Aggregates crystals | gamma ($\mu = 7$) | 4–50 | 6–15 | 0.1–20 |

### 3.2. Comparison with the Reference Model

The radiative transfer code LBLRTM is exploited for the generation of gaseous optical depths. The gas ODs are then, respectively, ingested by the DISORT routine and a Python script implementing the MAMA methodology. For comparison, a set of simulations based on the scaling method proposed by Chou et al. (1999) [30] is also performed for each simulated condition. Radiance simulations are run at 0.01 cm$^{-1}$ for both MAMA and DISORT. The number of streams used in the DISORT simulations is 18, which is considered a reasonable trade-off between accuracy and computational time. Successively, high

spectral resolution radiances are convolved to the nominal FORUM Sounding Instrument resolution of 0.5 cm$^{-1}$ by using a sinc function. To evaluate the accuracy of the approximate methodology, we compare the FORUM-like radiance values obtained from the MAMA solution to those obtained using the discrete ordinate method. Specifically, the differences are computed as shown in Equation (33):

$$\Delta I = I_M - I_r \tag{33}$$

where $I_M$ is the radiance obtained from the MAMA methodology (or the Chou scaling method) and $I_r$ is that from the reference radiative transfer model. The differences between the computed values ($\Delta I$) are compared to the goal noise equivalent spectral radiance (NESR) of the FORUM mission. This approach provides a metric to determine whether the differences $\Delta I$ are within acceptable limits. Specifically, the FORUM goal NESR is 0.4 mW/(m$^2$cm$^{-1}$sr) within the 200–800 cm$^{-1}$ spectral region and 1.0 mW/(m$^2$cm$^{-1}$sr) outside of it (https://esamultimedia.esa.int/docs/EarthObservation/FORUM_MRD_v2.0 _091220_issued.pdf, accessed on 11 July 2023).

### 3.2.1. Liquid Water Cloud

Low-level liquid water cloud spectra for the three climatological profiles using multiple values of cloud top height, effective radii, and optical depth are computed. A first example of comparison comprising the full spectrum is reported in Figure 3 for a mid-latitude cloud layer with optical depth of 10 and altitude placed at 3.0 km. The difference with respect to the reference code ($\Delta I$) is calculated as shown in Equation (33) for both the MAMA solution and the Chou scaling method. The shaded grey area indicates the range of the goal FORUM NESR.

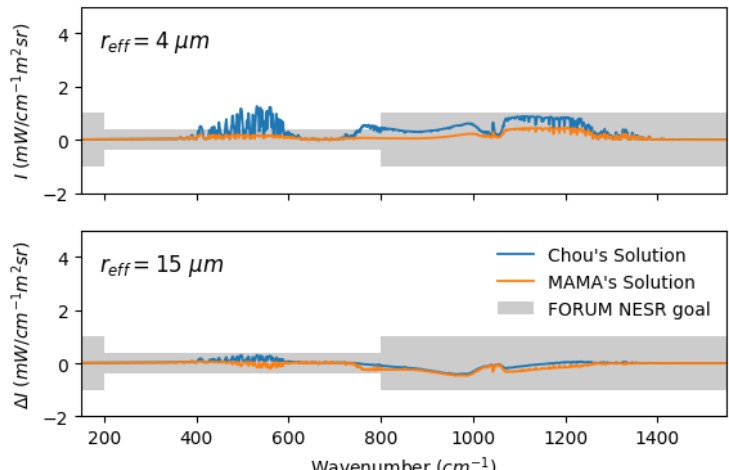

**Figure 3.** Top panel: radiance differences between approximate solutions (Chou's scaling method in blue and MAMA solution in orange) and the reference approach for a mid latitude water cloud with $OD = 10$, $r_{eff} = 4$ µm and cloud top at 3 km. Bottom panel: the same as upper panel, but for an effective radius of 15 µm. The range of values of the FORUM NESR are highlighted by a grey shaded area.

Both solutions show a $\Delta I$ nearly zero at small and large wavenumbers, i.e., below 350 cm$^{-1}$ and above 1400 cm$^{-1}$, which are spectral intervals affected by high atmospheric absorption caused by water vapor. Similarly, the strong absorption due to the $CO_2$ band around 667 cm$^{-1}$ completely masks the cloud effects on the top-of-atmosphere radiance. Figure 3 highlights that the MAMA methodology accurately simulates the considered water clouds at any wavenumber in the longwave part of the spectrum. Notably, the MAMA method achieves accurate results also for small effective radii of the PSD, which is typically challenging for scaling methods [33] due to the enhanced multiple-scattering effects.

To provide a comprehensive assessment of the accuracy of the MAMA method compared to the reference solution, a set of results is reported in Figures 4 and 5. The multiple panels of the figures show the MAMA-DISORT radiance differences evaluated at two specific wavenumbers representative of the MIR (1203 cm$^{-1}$) and the FIR (531 cm$^{-1}$) for various atmospheric and liquid water cloud conditions. The selected wavenumbers constitute the worst cases in terms of radiance differences in the mid- and far- infrared regions. The values are displayed as a function of the cloud OD and effective radius to assess the dependence of the results on these parameters. Note that, for both Figures 4 and 5, the panel accounting for liquid water clouds at high latitudes and with cloud top at 3 km is not plotted due to the rare occurrence of these conditions.

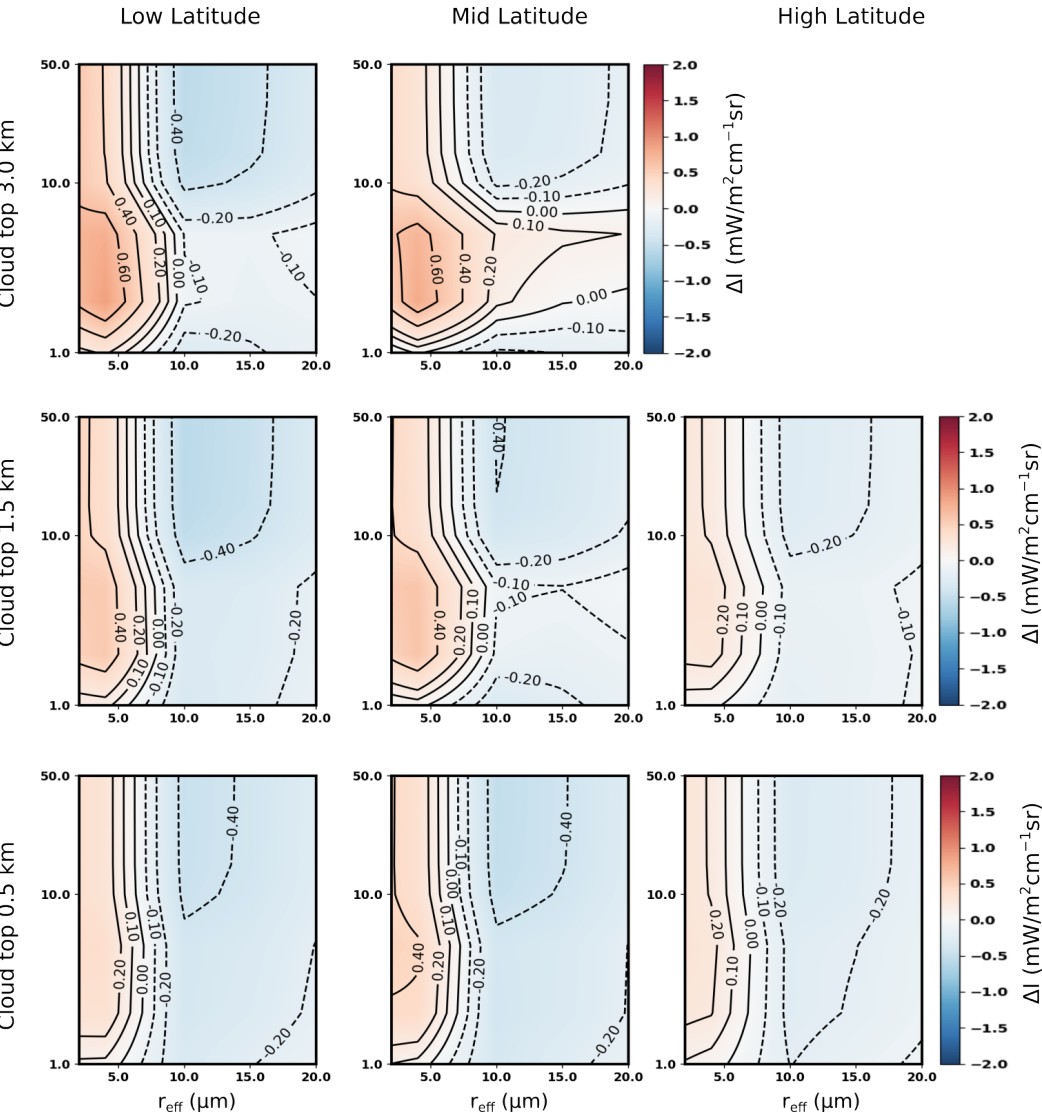

**Figure 4.** Radiance differences ($\Delta I$) between the MAMA and the reference solution at 1203 cm$^{-1}$ (MIR), for multiple liquid water clouds (varying ODs and $r_{\mathrm{eff}}$) and atmospheric conditions (low, mid and high latitude). If present, the red and blue contour lines highlight the regions, where the differences values are above the goal FORUM NESR.

The contour lines in the figures show the $\Delta I$ values in units of mW/(m$^2$cm$^{-1}$sr). As a term of comparison, the FORUM NESR goal is 1.0 mW/(m$^2$cm$^{-1}$sr) at 1203 cm$^{-1}$ and 0.4 mW/(m$^2$cm$^{-1}$sr) at 531 cm$^{-1}$. The plots are meant to provide a detailed overview of the accuracy of the MAMA computational method for various atmospheric and liquid water cloud conditions.

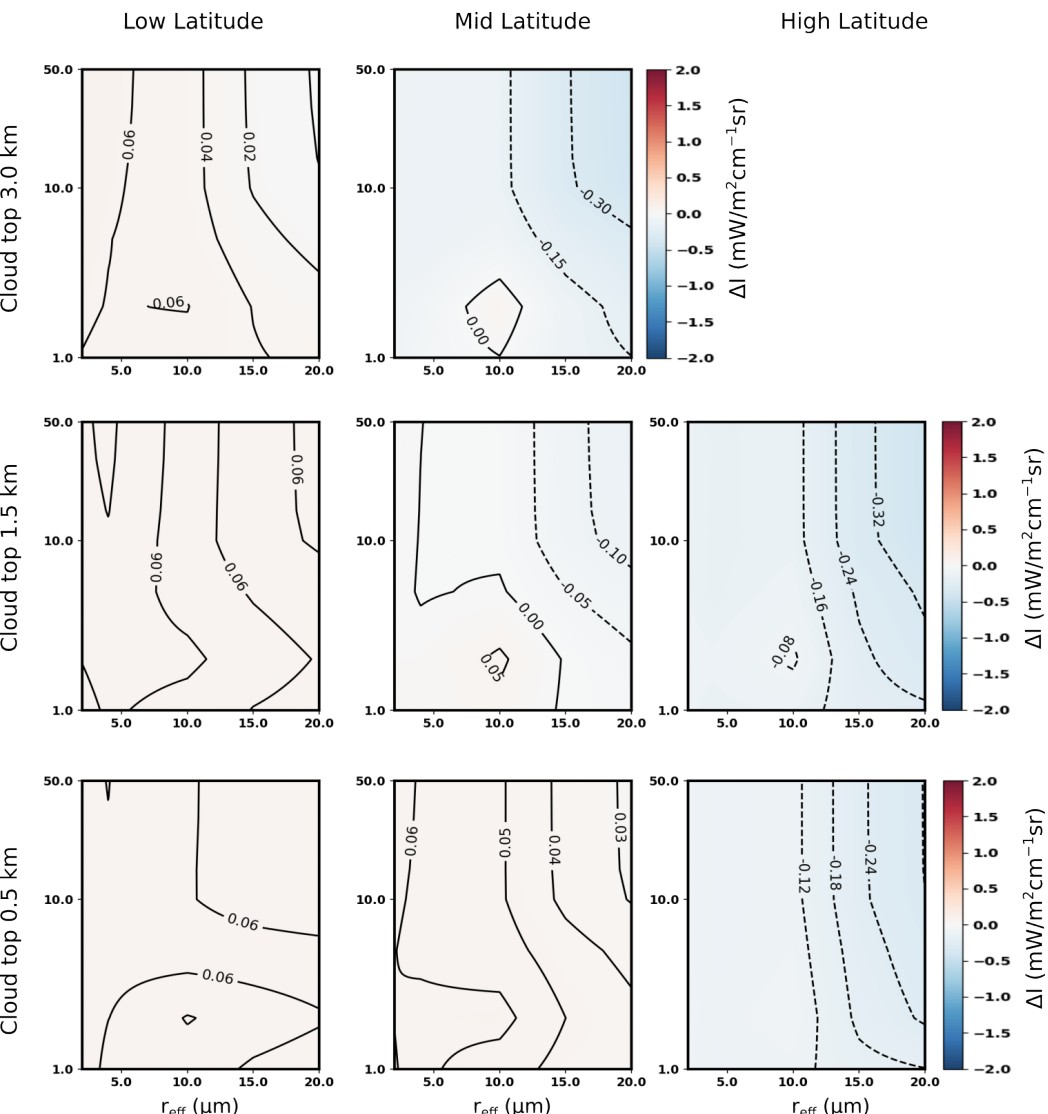

**Figure 5.** Same as Figure 4 but for radiance differences ($\Delta I$) at 531 cm$^{-1}$ (FIR).

Figures 4 and 5 highlight the effectiveness of the new methodology in computing top of the atmosphere radiances in the presence of water clouds. The MAMA solution is capable of obtaining accurate simulations (with respect to the FORUM NESR goal) for all the atmospheric and cloud conditions considered, which includes very extreme cases, such as high optical depths and small effective radii (and vice versa). This holds over the entire spectral region considered (FIR and MIR) and also when small effective radii of the PSD are assumed. For $r_{eff}$ larger than 10 µm, the MAMA solution produces a slight underestimation of the upwelling radiance, which is anyway less than 0.4 mW/(m$^2$cm$^{-1}$sr) and thus smaller than the FORUM NESR.

At FIR wavenumbers and in a tropical scenario (first column of Figure 5), the radiance differences are close to zero for all the studied cases also because of the attenuating effect of precipitable water vapor (PWV) above the cloud layers, which is related to the upper atmospheric layer transmissivity. The higher the PWV above the cloud top level, the smaller the radiance difference. This masking effect becomes particularly effective for PWV values larger than about 4 mm [33], and it is the reason why radiance differences (slightly) increase with the increasing cloud top for a fixed latitude, as well as with increasing latitude for fixed cloud top height.

### 3.2.2. Ice Cloud

The spectral radiance differences between simulations obtained using the MAMA methodology and the reference algorithm (DISORT) are evaluated also in the presence of ice clouds. Figure 6 illustrates a result for a full spectrum, where the top of the atmosphere radiance differences are plotted for a mid-latitude cirrus cloud (top altitude placed at 8 km) and for two different effective radii ($r_{\text{eff}} = 20$ and $r_{\text{eff}} = 30$ μm). The cloud OD is set to unity so that the multiple-scattering effects are maximized [37] and the quality of the scaling method can be tested in challenging configurations. The results show the excellent level of agreement of the MAMA approach with the reference solution over the entire spectral range considered. Specifically, the MAMA code performs significantly better than the Chou scaling method in reproducing the spectral radiance at FIR wavenumbers, where scattering effects are amplified by the minimum in ice absorption (with maximum effect at around 410 cm$^{-1}$).

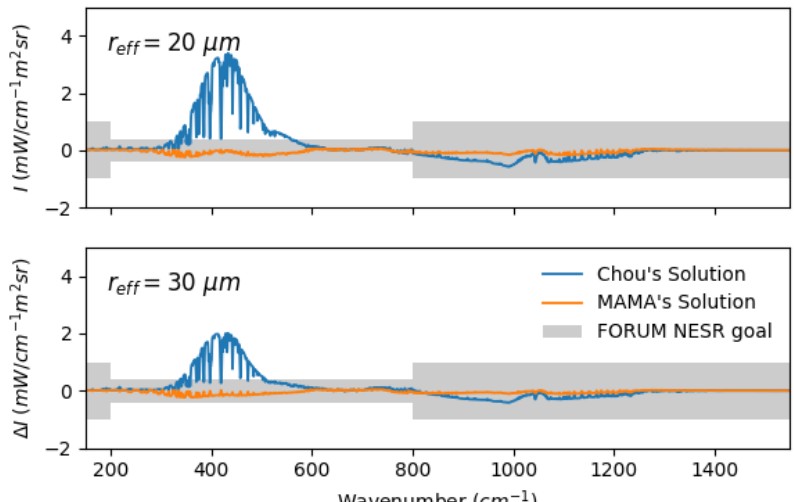

**Figure 6.** Top panel: radiance differences between the approximate solutions (Chou's scaling method in blue and MAMA in orange) and the reference algorithm in the presence of a mid latitude ice cloud with $OD = 1$, $r_{\text{eff}} = 20$ μm and top altitude at 8 km. Bottom panel: the same as above, but for an effective radius of 30 μm. The range of values of the FORUM NESR is highlighted by a grey shaded area.

The overall accuracy of the MAMA methodology for different atmospheric and ice cloud conditions is evaluated at two wavenumbers. The chosen wavenumbers are representative of the highest spectral discrepancies along the spectrum between the MAMA and the reference code solution in the MIR and FIR spectral regions considered. The multiple panels in Figures 7 and 8 show the $\Delta I$ values at 1203 cm$^{-1}$ and 410 cm$^{-1}$ for three typical atmospheric conditions (low, mid and high latitudes), and for varying cloud altitude, OD, and effective radius. A logarithmic scale is used on the *OD* axis to highlight the MAMA accuracy at small values of cloud optical depths (i.e., OD $\leq$ 2) since the study of cirrus clouds is one of the targets of the FORUM mission. Note that ice clouds with a top altitude higher than 12 and 8 km are not considered for mid-latitude and high-latitude conditions, respectively, due to their rare occurrence. Similarly, low-level ice clouds at low latitudes are ignored; in this case, the high concentration of water vapor in the upper layers of the tropical atmosphere makes the FIR channels almost opaque to low–medium level clouds.

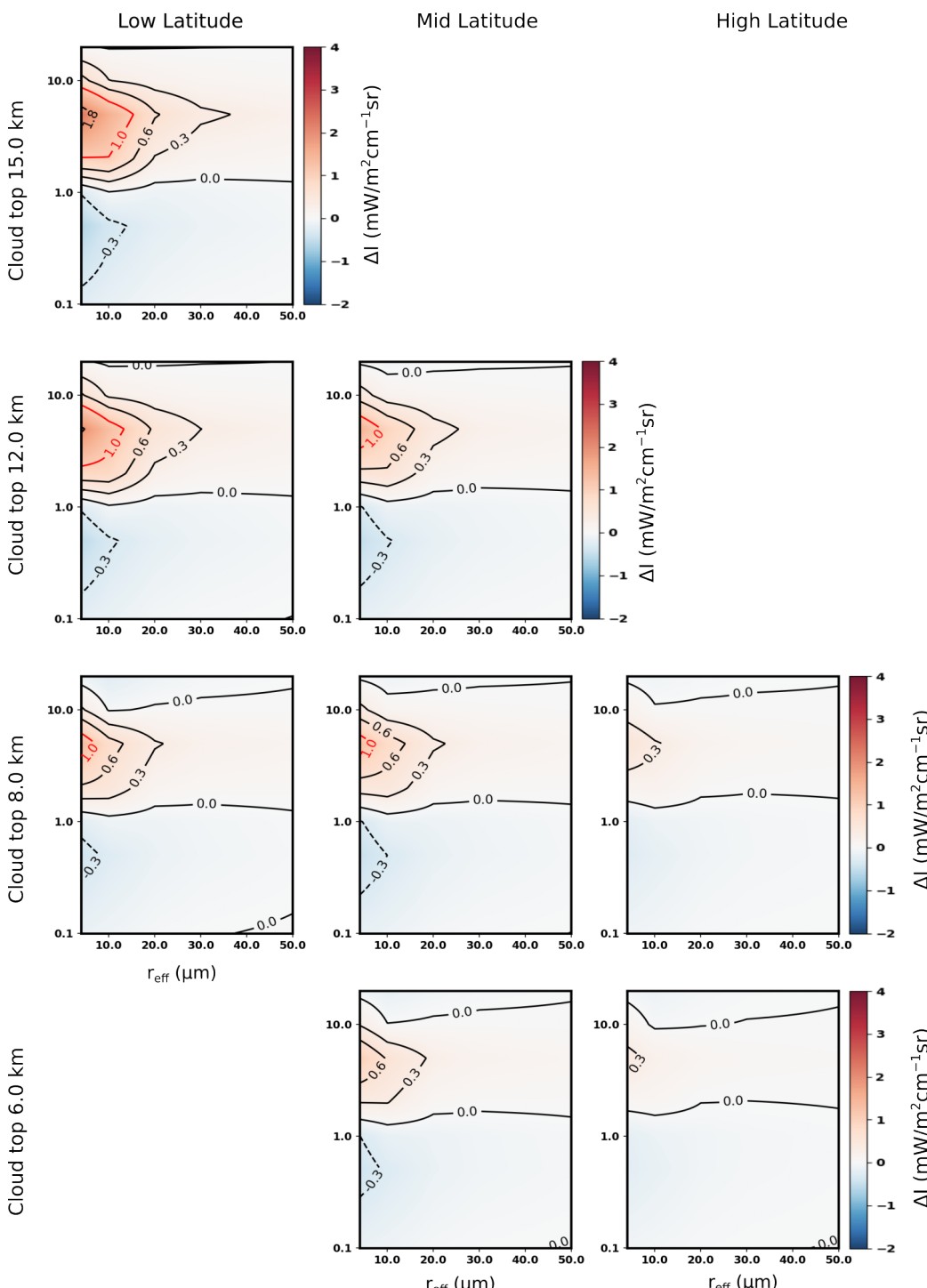

**Figure 7.** Radiance differences ($\Delta I$, contour) between the MAMA and the full physics solution at 1203 cm$^{-1}$ (MIR), for multiple ice clouds (varying ODs and $r_{eff}$) and atmospheric conditions (low, mid and high latitude). If present, the red and blue contour lines indicate the regions where the differences values are above the goal FORUM NESR.

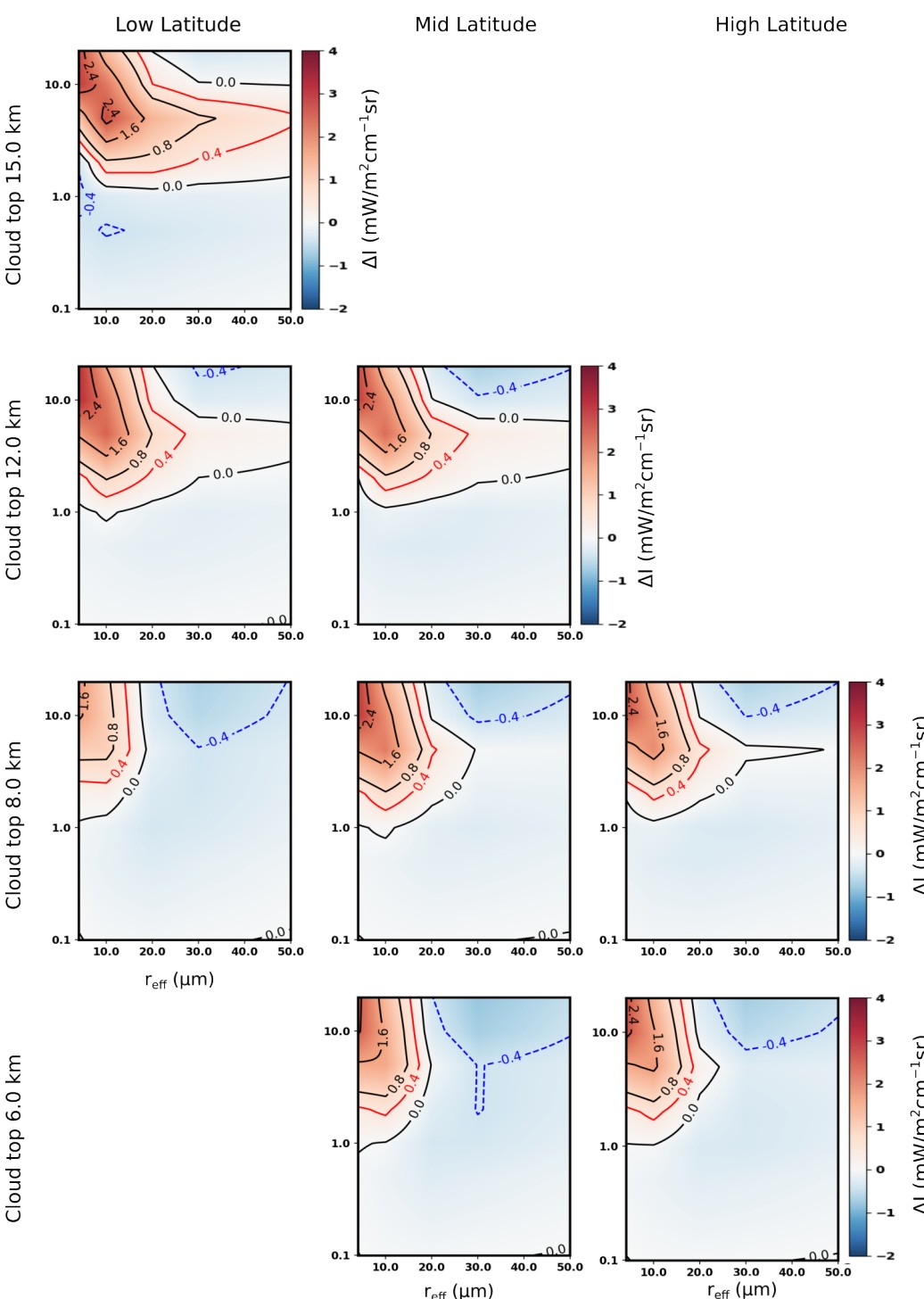

**Figure 8.** Radiance differences ($\Delta I$, contour) as in Figure 7 but for values at 410 cm$^{-1}$ (FIR).

Figure 7 displays the radiance differences at MIR (1203 cm$^{-1}$). The computed $\Delta I$ values are small all over the entire set of scenarios considered, with the exception of cloud cases characterized by high altitudes, small effective radii ($r_{\text{eff}} < 15$ μm) and *OD* around 5. The predominantly pale color in most of the panels of the figure indicates that the difference between the two solutions is lower than the FORUM NESR goal at that particular wavenumber. The results confirm the reliability of the MAMA methodology at mid-infrared (MIR) wavelengths for the simulation of both thick ice clouds and cirrus clouds. It is worth noting that the monthly mean effective radius derived from MODIS AQUA L3 products

for analogous scenarios is approximately 30 micrometers in the presence of ice clouds (including cirrus) [45].

Simulations at 410 cm$^{-1}$ are reported in Figure 8 which shows that the MAMA solution deviates from the reference code only for OD greater than 5 and small effective radii ($r_{\text{eff}} < 15 \, \mu$m). For optically thin cirrus clouds ($OD < 2$), areas with $\Delta I$ values lower than FORUM NESR are observed almost independently of the assumed effective dimension of the PSD. As the cloud OD increases, the accuracy of the solution gets worse, and the synthetic radiances are slightly overestimated in the case of small effective radii and underestimated for large ones. Nevertheless, the overestimation never exceeds 2 mW/m$^2$cm$^{-1}$sr. The values reported in Figure 8 represent the largest discrepancies in terms of radiance across the entire FORUM spectrum.

The MAMA method is implemented for different particles habits available in the reference single-scattering single-particle dataset [6]. For comparison an example of the results obtained for other habits is reported in Figure 9. The upper panels refer to $\Delta I$ obtained for a tropical atmospheric scenario and ice clouds with a top height of 12 km when the 1203 cm$^{-1}$ wavenumber is considered. The simulations are performed considering plate (left panel) and solid column (right panel) ice crystals. The lower panels show results for the wavenumber at 410 cm$^{-1}$. The results are in accordance with the simulations obtained when in the presence of column aggregates particles and show the capability of the MAMA methodology to accurately simulate spectral radiances for optically thin clouds ($OD < 2$) also at FIR spectral bands.

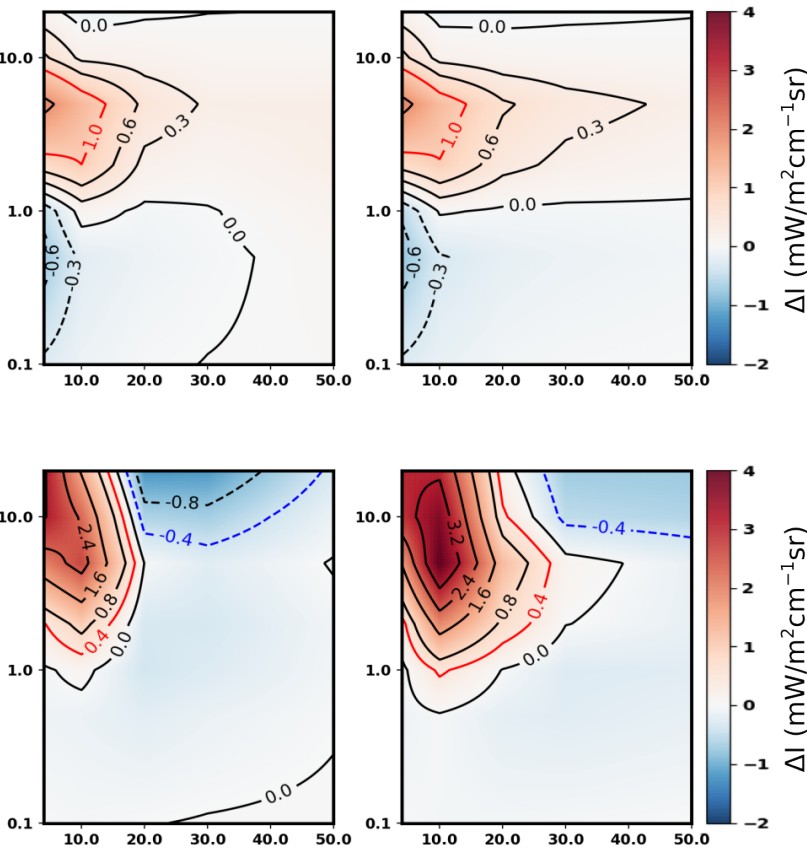

**Figure 9.** Top panels: Radiance differences ($\Delta I$, contour) between the MAMA and the full-physics solution at 1203 cm$^{-1}$ (MIR), for ice clouds characterized by plates crystals (left panel) and solid columns (right panel) and varying ODs and $r_{\text{eff}}$. The cloud is placed at 12 km of altitude and in tropical conditions (Low latitude). Red and blue contour lines indicate the regions where the differences values are above the goal FORUM NESR. Bottom panels: same as above but at 410 cm$^{-1}$ (FIR).

## 4. Conclusions

In this work, a new radiative transfer methodology for spectral infrared simulations in the presence of cloud is presented. The method is called MAMA, and its computational algorithm takes inspiration from scaling methods; however, it is tailored to compute the up-welling radiance in the presence of any type of scattering layers. Scaling methods have become very popular in the radiative transfer modeling community due to their theoretical simplicity and ease of implementation. In general, these methods (e.g., [30]) allow clear sky radiative transfer models to simulate cloudy conditions, maintaining the same computational structure by approximating the scattering processes with scaled absorption properties. Since the scaling approach is usually based on simplistic assumptions concerning the downwelling radiation, correction terms are often introduced (as in [31]), which slightly complicate the computational algorithm's structure. While scaling methodologies were developed for broadband longwave irradiance computations, they are also effectively applied for calculating spectral infrared radiance [46,47]. The advantage in this sense relies on the speed of computations, which, being of the order of 0.x seconds on a standard personal computer for a full-infrared spectrum, results in being smaller by about one order of magnitude with respect to a two-stream model (i.e., [29]). The MAMA methodology is built to be as fast as scaling methods in terms of computational time, easy to implement and upgrade in terms of radiative properties, and accurate all over the longwave spectral range from far- to mid-infrared regions. The MAMA algorithm also has the advantage of being based on the fundamental optical properties of the layers, and thus it allows to retain an explicit treatment of the cloud micro-physics, which is usually lost in hyper-fast methodologies, such as those given in [26].

The solution is based on the separation of the multiple-scattering term in two parts (back- and forward-scattering contributions), which are described as the product of an effective average ambient radiation ($\langle I_u \rangle$ and $\langle I_d \rangle$), and a new parameter called the angular back-scattering coefficient, $c(\mu)$. The physical meaning of $c(\mu)$ is related to the radiation coming from a hemisphere and back-scattered into a specific direction, such as the one defined by the observer ($\mu$). The back-scattering coefficient is easily computed from the phase function of the particle size distribution of the layer; it is a fundamental property of the layer itself, meaning that it defines how the layer interacts with radiation in terms of scattering processes. The variability of $c(\mu)$ with the observational angle is smooth, as shown in Appendix B, which makes the computed values representative of a set of angles and allows its application for radiative transfer computations.

The upward effective ambient radiation $\langle I_u \rangle$ is calculated assuming linearity in the $\mu$ dependency of the radiance $I(\mu)$ within the considered hemisphere. The limit value of $I(\mu)$ for $\mu = 0$ is estimated, within a scattering layer, using a Chou-like scaling methodology. On the other hand, the value of the downward effective ambient radiance $\langle I_d \rangle$ is approximated as the downward radiance computed using the Chou solution for an effective angle $\tilde{\mu}$. The resulting Equation (22) is a Schwarzschild-like equation based on fundamental parameters easily derivable from the optical properties databases of the scattering layers.

The MAMA accuracy is assessed by comparing the computed upwelling radiance with those derived using a reference solution (DISORT), which is taken as the truth within the goal noise-equivalent spectral radiance of the future FORUM (Far-infrared Outgoing Radiation Understanding and Monitoring) sensor. FORUM is the 9th ESA Earth Explorer and will observe both the far- and mid-infrared parts of the spectrum at about 0.5 cm$^{-1}$ unapodized resolution. As for water clouds, the MAMA code accurately simulates far- and mid-infrared spectral radiances for all the combinations of optical depths, effective dimensions and cloud altitudes accounted for in the simulations. Radiance spectra in the presence of ice clouds are mostly accurately simulated with the exception of very uncommon cases, which accounts for small crystal dimensions and large optical depth and only at far-infrared wavelengths. In conclusion, the proposed methodology is suitable for the computation of radiance spectra at high resolution at all mid-infrared wavenumbers and in the case of any cloud conditions. This makes it an appropriate tool for the

analysis of observations from current infrared sounders, such as the Infrared Atmospheric Sounding Interferometer-IASI (https://www.eumetsat.int/iasi) or the Cross-track Infrared Sounder-CrIS (https://www.nesdis.noaa.gov/current-satellite-missions/currently-flying/joint-polar-satellite-system/cross-track-infrared), among others. In the far-infrared part of the spectrum, the MAMA algorithm shows the largest improvements with respect to previous scaling methods [33,34]. Despite the challenging conditions due to enhanced scattering processes, MAMA code accurately simulates far-infrared radiances in the presence of ice clouds with optical depth less than 2 and for any atmospheric condition. The MAMA computational performance results in terms of speed, its accuracy at the far-infrared region and its readiness for implementation as well as its easiness in updating, make it suitable to be applied to the future measurements of the FORUM mission.

**Author Contributions:** Conceptualization, T.M.; Methodology, M.M. and T.M.; Software, M.M.; Formal analysis, M.M.; Writing—original draft, M.M.; Writing—review & editing, T.M.; Supervision, T.M.; Funding acquisition, T.M. All authors have read and agreed to the published version of the manuscript.

**Funding:** This research received no external funding.

**Data Availability Statement:** Not avaliable.

**Acknowledgments:** The authors acknowledge the Italian Space Agency (ASI) for the financial support provided through the FORUM-SCIENZA and, the recently accepted, FIT-FORUM projects dedicated to the development of software tools for the analysis of FORUM observations. These projects, although not directly funding the research presented, have created the necessary conditions for the authors to devote themselves to the development of the MAMA methodology.

**Conflicts of Interest:** The authors declare no conflict of interest.

## Appendix A. Derivation of $\overrightarrow{I}$

To obtain a suitable form for $\overrightarrow{I}$, we can start considering the following scenario. Let us place in a plane-parallel atmosphere a thin cloud layer, which is capable of absorbing, emitting and scattering the radiation. Assuming plane-parallel conditions, the equation governing the radiative transfer for the radiance parallel to the surface inside the cloud is written as

$$\frac{d\overrightarrow{I}(\tau,\tau',\mu=0)}{d\tau'} = \overrightarrow{I}(\tau,\tau',\mu=0) - [1-\tilde{\omega}(\tau)]B(\tau) - \frac{\tilde{\omega}(\tau)}{2}\int_{-1}^{1}d\mu' I(\tau,\mu')P(\mu',\mu=0) \tag{A1}$$

where $\tau$ indicates the optical depth along the vertical coordinate and $\tau'$ along the horizontal coordinate, i.e., the optical depth along the horizontal direction. The term $I(\tau,\mu')$ inside the integral is the ambient radiation at level $\tau$. This last term does not depend on $\tau'$, as we are considering a horizontal symmetry for this problem. Figure A1 shows a schematic representation of the horizontal radiance $\overrightarrow{I}$.

To estimate the solution of Equation (A1), simple assumptions are considered regarding the ambient radiation $I(\tau,\mu')$. Similar to what was performed by Chou et al. (1999) [30], it is assumed that the radiation coming from the upper hemisphere ($\mu' < 0$) is equal to the Planckian emission from the layer $B(\tau)$. On the other hand, the upward ambient radiation ($\mu' > 0$) is taken to be equal to the incident upward radiance $I(\mu=1)$. Thus, in summary, the ambient radiation within the cloud is here assumed:

$$I(\tau,\mu') = \begin{cases} B(\tau) & \text{if } \mu' < 0 \\ I(\mu=1) & \text{if } \mu' > 0 \end{cases} \tag{A2}$$

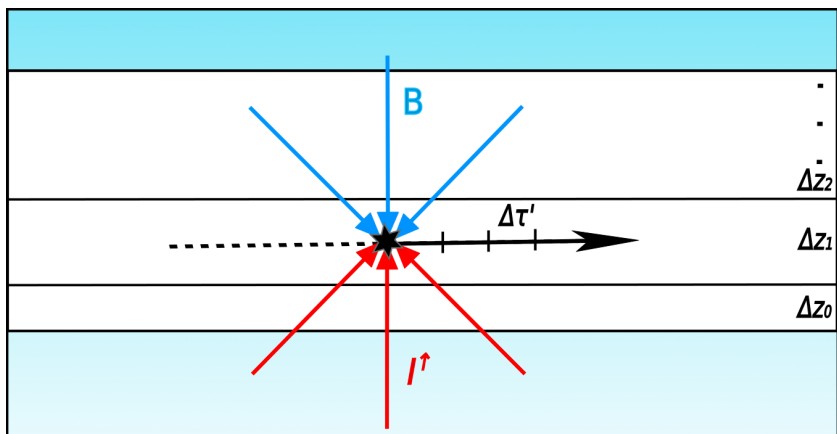

**Figure A1.** Schematic representation of the processes assumed to model the horizontal radiance $\overrightarrow{I}$ within a cloud layer.

Using the (A2) in Equation (A1), the following is obtained:

$$
\frac{d\overrightarrow{I}(\tau,\tau',\mu=0)}{d\tau'} = \overrightarrow{I}(\tau,\tau',\mu=0) - [1-\tilde{\omega}(\tau)]B(\tau) +
$$
$$
- B(\tau)\frac{\tilde{\omega}(\tau)}{2}\int_{-1}^{0} d\mu' P(\mu',\mu=0) - I(\mu=1)\frac{\tilde{\omega}(\tau)}{2}\int_{0}^{1} d\mu' P(\mu',\mu=0) \tag{A3}
$$

where the multiple-scattering term is split into two contributions, one for the downward and one for the upward ambient radiance, respectively. Using the definition for the directional back-scattering coefficient given in (4), and noting that $c(\mu=0)=0.5$, we can write Equation (A3) as

$$
\frac{d\overrightarrow{I}(\tau,\tau',\mu=0)}{d\tau'} = \overrightarrow{I}(\tau,\tau',\mu=0) - [1-\tilde{\omega}(\tau)]B(\tau) - B(\tau)\frac{\tilde{\omega}(\tau)}{2} - I(\mu=1)\frac{\tilde{\omega}(\tau)}{2} \tag{A4}
$$

which is solved by

$$
\overrightarrow{I}(\tau,\tau',\mu=0) = \overrightarrow{I}_0(\tau)e^{-\tau'} + \left[\left(1-\frac{\tilde{\omega}(\tau)}{2}\right)B(\tau) + \frac{\tilde{\omega}(\tau)}{2}I(\mu=1)\right]\left(1-e^{-\tau'}\right) \tag{A5}
$$

where $\overrightarrow{I}_0(\tau)$ is the boundary condition for the parallel radiance. Taking the limit for a large optical path $\tau'$ (the cloud extends along the horizontal direction), the solution simplifies to

$$
\overrightarrow{I}(\tau,\mu=0) = \left(1-\frac{\tilde{\omega}(\tau)}{2}\right)B(\tau) + \frac{\tilde{\omega}(\tau)}{2}I(\mu=1) \tag{A6}
$$

In this solution, the parallel radiance is composed of only two contributions, consisting of one from the emission from the cloud layer itself and the other from the scattered radiation.

**Appendix B. Optical Properties of Water and Ice Clouds**

An overview of the optical properties required by the MAMA code is provided. In particular, the two newly defined parameters, namely the angular back-scattering coefficient $c(\mu)$ and the gamma coefficient $\gamma(\mu)$, are discussed.

The angular back-scattering coefficient is defined in a similar way to what was performed for the back-scattering coefficient $b$, proposed by Chou [30], but for application to radiance instead of irradiance. The angular back-scattering coefficient physical meaning is related to the fraction of radiation back-scattered toward a specific direction indicated

by $\mu$ with respect to the hemispheric incident radiation. Its mathematical definition (4) is repeated here for convenience:

$$c(\mu) = \frac{1}{2} \int_{-1}^{0} d\mu' P(\mu', \mu) \tag{A7}$$

The spectral variation of two angular back-scattering coefficients for a couple of cases of water and ice clouds are reported in Figure A2 in the upper and lower panels, respectively. For comparison, the spectral variation of the $b$ coefficient is also plotted. Each panel of Figure A2 accounts for small and large effective radii. Results are computed for a nadir-looking geometry ($\mu = 1$). The angular back-scattering coefficient shows smooth variation with the wavenumber and value approaching 0.5 for small wavenumbers and particle dimensions. For large effective radii and large wavenumbers, the phase function presents a strong forward peak, and $c(\mu)$ reduces to values smaller than 0.1.

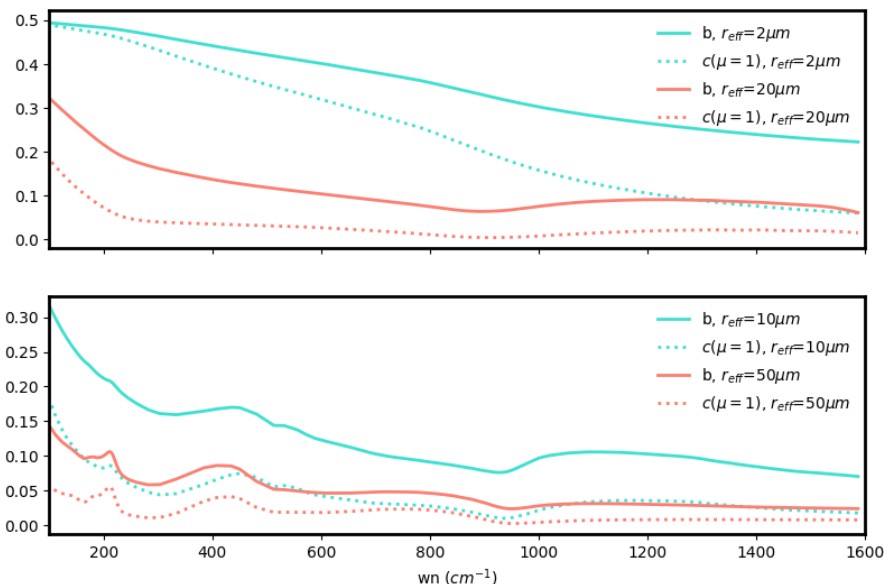

**Figure A2.** Upper panel: angular back-scattering $c(\mu = 1)$ and back-scattering $b$ coefficients for water clouds as a function of the wavenumber. Two effective radii are considered as reported in the legend. Lower panel: the same as upper panel but for ice clouds.

The gamma coefficient definition is similar to that for the asymmetry parameter $g$ except for the lower limit of integration over the zenith angle. In fact, in the case of $\gamma$, the integration is performed over the forward hemisphere only:

$$\gamma(\mu) = \frac{1}{2} \int_{0}^{1} d\mu' P(\mu', \mu) \mu' \tag{A8}$$

For nadir-looking observations ($\mu = 1$), the cosine of the angle between the entering and exiting direction $\theta$ in the scattering problem results in being equal to $\mu'$ as is easily derived from Equation (A9) relating the scattering angle to the zenith and azimuth of exiting and entering angles:

$$cos(\theta) = \mu\mu' + (1 - \mu^2)^{\frac{1}{2}}(1 - \mu'^2)^{\frac{1}{2}} cos(\phi' - \phi) \tag{A9}$$

In the case of highly forward peaked phase function (usually associated to limited back-scattering), the resulting gamma parameter's values are very close to those obtained for the asymmetry parameter $g$.

Figure A3 shows the spectral variation of $\gamma$ for both water and ice clouds. Each panel considers cloud PSDs representatives of small and large effective radii. Results

are computed for a nadir-looking geometry ($\mu = 1$). For comparison, the $g$ coefficients computed in the same cloud conditions are also reported in the figure. As expected, the $\gamma$ values are larger than $g$ values, and the spectral variations are smooth. The gamma parameter increases for large particle dimensions and large wavenumbers.

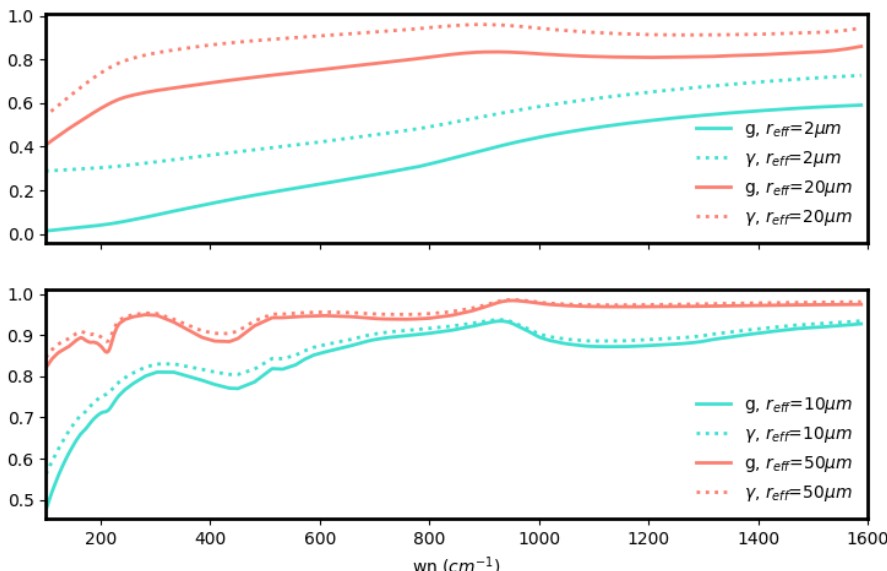

**Figure A3.** Upper panel: Values of the asymmetry parameter $g$ (solid line) and the gamma coefficient $\gamma(\mu = 1)$ (dotted line) of a water cloud as a function of the wavenumber. The values are reported for two different effective radii of the distribution. Lower panel: the same as above but for an ice cloud.

The main focus of this study is on the RT solution at nadir-looking geometry; thus, the cloud optical parameters (the angular back-scattering coefficient and the gamma parameter) are computed under the assumption of a fixed viewing angle $\mu = 1$. Nevertheless, it is noteworthy that these quantities exhibit a smooth variation with the viewing angle. This feature makes the computed properties representative not only of nadir-looking angles but also of small off-nadir angles. Figures A4 and A5 show the value of the angular back-scattering parameter $c$ as a function of the viewing angle for a liquid water cloud and an ice cloud, respectively. At FIR and MIR wavenumbers, the variation of the $c$ parameters is less than 2% for off-nadir angles smaller than $10°$.

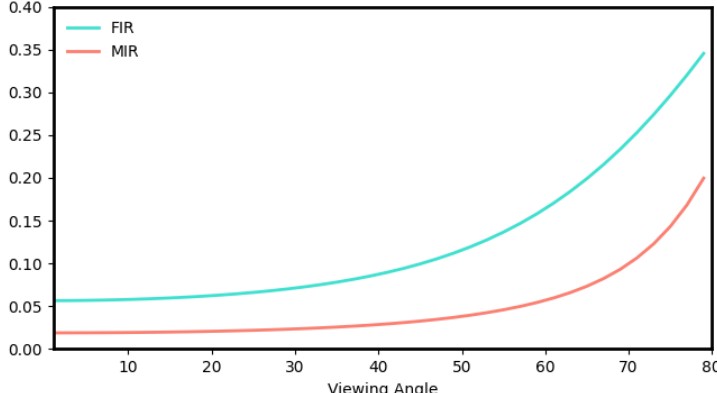

**Figure A4.** The angular back-scattering coefficient $c(\mu)$ as a function of the viewing angle (in degrees). A water cloud with an effective radius of 10 μm is considered. The blue and pink lines refer to wavenumbers at 531 (FIR) and 1203 (MIR) cm$^{-1}$, respectively.

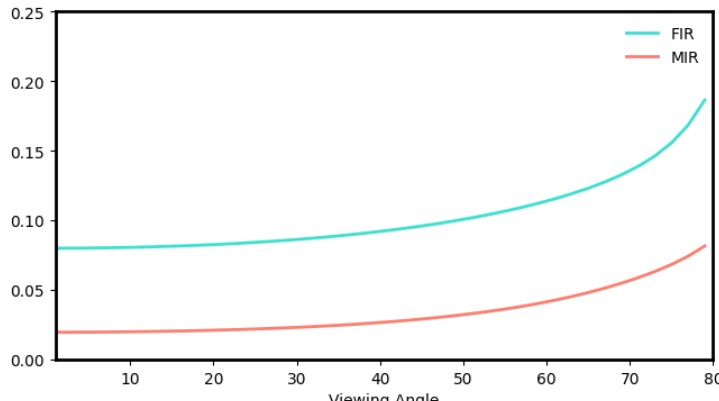

**Figure A5.** Angular back-scattering coefficient $c(\mu)$ as a function of the viewing angle (in degrees). An ice cloud with effective radius of 20 μm is considered. The blue and pink lines refer to wavenumbers at 410 (FIR) and 1203 (MIR) cm$^{-1}$, respectively.

**Appendix C. Solution for Off-Nadir Observation Angles**

The MAMA solution can be generalized for any observational angle. The extension of Equation (22) requires to account for explicit dependence on the cosine of the viewing angle, $\mu$. In this regard, Equation (8) becomes

$$
\begin{aligned}
\frac{1}{\mu}\frac{dI(\tau,\mu)}{d\tau} = I(\tau,\mu) - [1 - \tilde{\omega}(\tau)]B(\tau) + \\
- \tilde{\omega}(\tau)[\langle I_d(\tau,\mu)\rangle c(\mu) + \langle I_u(\tau,\mu)\rangle (1 - c(\mu))]
\end{aligned}
\tag{A10}
$$

where the mean downward and upward ambient radiations explicitly depend on $\mu$.

The derivation of these two quantities mirrors the one already proposed for the nadir-looking geometry. Specifically, the mean downward ambient radiation remains unchanged since its expression (Equation (20)) does not contain any directional term and it can thus be considered to be general. On the other hand, the mean upward ambient radiation exploited in the MAMA solution (Equations (12) and (15)) was written to account for the specific upward radiation along the vertical direction. For a generalization of the mean upward radiation, it is thus necessary to modify the linear relation assumed in Equation (9). A possible solution for the ambient radiation is provided by the following relation:

$$
I(\tau,\mu',\mu) = \frac{I(\tau,\mu) - \overrightarrow{I}(\tau)}{\mu}\cdot\mu' + \overrightarrow{I}(\tau), \qquad \mu' > 0
\tag{A11}
$$

where the limit case for $\mu' = 1$ is linearly extrapolated.

Equation (A11) can be used to derive a new expression for the mean ambient radiation, and from this, a general solution of the MAMA algorithm for any observational angle is obtained. The detailed derivation of the general solution and the assessment of the accuracy of the code at any observational angle are not provided in this work and require further extended investigations and computations.

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
