# Peer review of "The MAMA Algorithm for Fast Computations of Upwelling Far- and Mid-Infrared Radiances in the Presence of Clouds"

_remotesensing, doi:10.3390/rs15184454_

Round 1

Reviewer 1 Report

The manuscript describes a new approximation to the multiple scattering radiative transfer equation, appropriate for thermal infrared regimes in Earth's atmosphere. The authors demonstrate the accuracy of the approximation by comparing to a full multiple scattering calculation using a "reference" scattering RT solver, DISORT. The results are presented in the context of the upcoming ESA FORUM mission.

The manuscript is well motivated and appropriate for the journal. I have only minor issues. Specific general issues:

At Line 213, the authors state: "However, it is important to highlight that this methodology can also be extended to encompass other viewing angles." after stating that the focus will be on the nadir geometry. I think the manuscript would be more impactful if this was revisited later; is there a readily derivable form of equation 24 that has an explicit dependence on mu? That could be added as an additional appendix, so it does not disrupt the narrative in the existing manuscript.

I recommend stating the plane parallel assumption more clearly up front, for example in the abstract, just to make it extremely clear to the reader in what regimes the new methods are valid.

The data displayed in the contour figures (#4, 5, 7, 8, 9) is not presented well. I understand the focus on the FORUM spectral noise specification, but plotting the data as white when it is within the spectral noise envelope makes these figures far less informative. Since FORUM is a hyperspectral instrument, biases smaller than the noise level (at a single spectral channel) can still significant, for retrievals that use the full spectral measurement. In addition, readers may be interested to see amplitude of the biases from a more general perspective, e.g. for other IR instrument concepts.

Instead of replacing the low amplitude regions with white, make the colors display according to the red/blue color map values. The FORUM NESR is already specified by a red/blue color contour, which is effective enough to illustrating the regions where the errors are small relative to FORUM.

Finally, the color bar range appears to be set to (-2,7), while the largest absolute value, if I am reading it correctly, is +4.0 in Figure 9. There is no reason for the upper range to go to +7, the range should be set to (-2,4).

In the equations, it is standard for characters that represent variables to be italicized, and other characters (such as units or descriptive subscripts). For example, if the document is typeset in LaTeX, I use \mathrm make this distinction, e.g. D_eff should be written in LaTeX as $D_\mathrm{eff}$, and wavenumber units are $\mathrm{cm}^{-1}$, etc.

Specific minor issues:

In the manuscript, LBLRTM and LBLDIS are written with lower case b (LbLRTM, LbLDIS) - I would recommend capitalizing the B to make it consistent with other literature.

Line 203: I think the claim should be that I_u and I_d are proportial to (not equal to) I_u and B(tau) (e.g., use \propto instead of \sim?)

Line 205: Here the authors describe that the Chou approximation leads to systematic high biases in the TOA upwelling radiance. Since it relies on hemisphere-averaged quanitity, then the bias should depend on the viewing zenith angle. In other words, I think this statement must apply to only small zenith angles, and the bias would change sign at high zenith angles.

Line 391 - somewhere (in this section or nearby) - need better description of the reference calculation. what was the number of streams used in DISORT? This is critical since DISORT could be run in a low number of streams, as another alternative 'fast' approximation.

What spectral resolution was computed? Is everything convolved to a FORUM ILS model before comparison, or is everything done on a LBL-like (or quasi-LBL) spectral resolution?

Line 448 - I would not use the phrase "full-physics solution" to describe the DISORT modeled radiances. Instead call this "reference algorithm" or something similar. MAMA is still a "physics algorithm", just an approximate one - would you agree? The other case is something like PCRTM or a machine learning approach where the radiance is computed from a statistical algorithm, instead of derived from physical equations.

There are some typos and unclear or incorrect english phrasing (but overall I would say it is minor.) I urge the authors to give another re-reading and use a spell checker. Here is a list of suggestions and grammatical/spelling corrections that I noticed during reviewing:

Line 11 - I would not use the verb 'retrieved' here, this is not discussing a geophysical property retrieval. Instead say 'can be calculated from any...."

Line 14: recommend using the more precise term "thermal infrared" instead of "longwave"

Line 64 "as long as the longwave region is considered" -> "in the thermal infrared"

Line 77: some of these algorithms are limited to the plane parallel assumption, correct? In other words, I do not think it is correct to say they solve the radiative transfer equation in any conditions.

Line 109 "This allows to obtain an extremely simple solution..." - the phrasing is unclear or incorrect english, suggest: "These approximations yield an extremely simple solution...". Similar unclear phrasing at line 259 could be improved, 

Line 115 tentative what? (I think a word is missing.)

Line 129 "...fundamental properties of the mean..." - this does not make sense, are there words missing?

Line 132 - "doesn't" - do not use contractions

Line 145 - "toa" -> "TOA" (this is an acronym)

Line 146, 274 "Plank" -> "Planck"

Figure 1 "angfular" -> "angular"

Line 258 "not-scattering" -> "non-scattering"

Line 332 "Caliop" -> "CALIOP" (this is an acronym)

Figure 8 "MIR" -> "FIR"

Figure 9: Are the bottom panels from the FIR wavenumber?

Line 508: "Despite being thought for broadband longwave irradiance computations, the scaling methodologies are also efficiently applied to simulate spectral infrared radiance"

I found this sentence unclear, suggest:

"While scaling methodologies were developed for broadband longwave irradiance computations, they are also effectively applied for calculating spectral infrared radiance."

References: #38 is missing

Reviewer 2 Report

The work entitled "The MAMA algorithm for fast computations of upwelling far and mid infrared radiance in presence of clouds" discusses an approximate form of the radiative transfer equation for calculating upward infrared radiances in the presence of thin clouds, where the multi-scattering term is approximated; this is worthy both for minimising the computing time, also in view of the future FORUM mission. 

The paper itself is very well written, it presents a thorough derivation of the RT equation, showing and explaining the reasoning behind the approximation of the scattering term, and the results show a good agreement against the benchmark method DISORT 

The topic of the paper is of relevance to the scientific community of the Special Issue, and the work will definitely contribute to the increase of knowledge in the field as well as to provide with new scientific tools.

I have no major concern, regarding the paper, except that it extensively relies on the nadir assumption: are we really sure that it would be straightforward for other viewing angles? Could you please elaborate more on this point, highlighting potential limits?

Please also find below few minor points, mainly typos:

line 268: "not-scattering" --> "non-scattering"

line 274: "Plank" --> "Planck"

line 437:  m2 -->  m^2 

line 458: "amplplified" --> "amplified"

line 477: remove "when"

line 481: remove "that the"

line 513: "two streams" --> "two-stream"

line 514: "ease" --> "easy"

line 549: "wavenumbes" --> "wavenumbers"

- Throughout the paper: "in presence of" -> "in the presence of"

- Please explain in the main text why some plots are missing in Figs. 4-8

English is fine except few typos
